# Current Advances, Research Needs and Gaps in Mycotoxins Biomonitoring under the HBM4EU—Lessons Learned and Future Trends

**DOI:** 10.3390/toxins14120826

**Published:** 2022-11-24

**Authors:** Paula Alvito, Ricardo Manuel Assunção, Lola Bajard, Carla Martins, Marcel J. B. Mengelers, Hans Mol, Sónia Namorado, Annick D. van den Brand, Elsa Vasco, Susana Viegas, Maria João Silva

**Affiliations:** 1National Institute of Health Dr. Ricardo Jorge (INSA), 1649-016 Lisboa, Portugal; 2Centre for Environmental and Marine Studies (CESAM), University of Aveiro, Campus Universitário de Santiago, 3810-193 Aveiro, Portugal; 3IUEM, Instituto Universitário Egas Moniz, Egas Moniz-Cooperativa de Ensino Superior, CRL, Campus Universitário—Quinta da Granja, Monte da Caparica, 2829-511 Caparica, Portugal; 4RECETOX, Faculty of Science, Masaryk University, Kotlarska 2, 611 37 Brno, Czech Republic; 5NOVA National School of Public Health, NOVA University of Lisbon, 1600-560 Lisbon, Portugal; 6Comprehensive Health Research Center, CHRC, 1600-560 Lisbon, Portugal; 7National Institute for Public Health and the Environment (RIVM), 3720 BA Bilthoven, The Netherlands; 8Wageningen Food Safety Research (WFSR), Part of Wageningen University & Research, 6708 WB Wageningen, The Netherlands; 9ToxOmics—NOVA Medical School, NOVA University of Lisbon, 1150-082 Lisboa, Portugal

**Keywords:** human biomonitoring, risk assessment, HBM4EU, human health, mycotoxins exposure, deoxynivalenol (DON), fumonisin B_1_ (FB_1_)

## Abstract

Mycotoxins are natural metabolites produced by fungi that contaminate food and feed worldwide. They can pose a threat to human and animal health, mainly causing chronic effects, e.g., immunotoxic and carcinogenic. Due to climate change, an increase in European population exposure to mycotoxins is expected to occur, raising public health concerns. This urges us to assess the current human exposure to mycotoxins in Europe to allow monitoring exposure and prevent future health impacts. The mycotoxins deoxynivalenol (DON) and fumonisin B_1_ (FB_1_) were considered as priority substances to be studied within the European Human Biomonitoring Initiative (HBM4EU) to generate knowledge on internal exposure and their potential health impacts. Several policy questions were addressed concerning hazard characterization, exposure and risk assessment. The present article presents the current advances attained under the HBM4EU, research needs and gaps. Overall, the knowledge on the European population risk from exposure to DON was improved by using new harmonised data and a newly derived reference value. In addition, mechanistic information on FB_1_ was, for the first time, organized into an adverse outcome pathway for a congenital anomaly. It is expected that this knowledge will support policy making and contribute to driving new Human Biomonitoring (HBM) studies on mycotoxin exposure in Europe.

## 1. Introduction

### 1.1. Human Exposure to Mycotoxins

Mycotoxins are secondary fungal metabolites often found as natural contaminants in agricultural commodities all over the world. They are closely associated with the consumption of crops previously contaminated with mycotoxigenic fungi, particularly cereals, although they also appear in fruits, vegetables, and animal products, including meat, dairy, and eggs [1,2]. Generally, mycotoxins are chemically and thermally stable compounds, surviving storage and most production process [3]. Their occurrence in such a variety of food and derived products is associated with human (and animal) exposure to mycotoxins on a daily basis, posing a risk to human and animal health. Food is the main source of exposure, and ingestion is the main route of exposure, to mycotoxins. In addition, inhalation and dermal routes can also contribute to exposure in certain occupational settings, mainly during tasks involving high exposure to organic dust (e.g., storage work, loading, handling, or milling contaminated grains, waste and feed) because airborne dust, spores, and hyphae fragments can act as carriers of mycotoxins to the lungs [4,5]. Currently, the main human and animal health burdens of mycotoxin exposure are related to chronic toxicity, such as carcinogenic, teratogenic, immunotoxic, nephrotoxic, and endocrine-disrupting effects. As such, it is crucial that the human exposure to mycotoxins is unraveled, which can be achieved either by determining mycotoxin occurrence in food products or by performing epidemiological and human biomonitoring studies, including the measurement of mycotoxins or their metabolites in human biological specimens. The latter, known as the direct approach, gives reliable insights into the internal dose achieved in the organism upon mycotoxin exposure.

The European Union has set maximum concentrations for mycotoxins in a range of foodstuffs to protect human health [6]. Besides these regulated mycotoxins, several other non-regulated toxins, such as modified and emerging mycotoxins, have been also reported in various foodstuffs [7,8,9]. Furthermore, co-exposure to several mycotoxins has been commonly found. However, their combined effect on human health needs to be considered and taken on board at a regulatory level when defining maximum concentrations [10]. In addition, mycotoxins are still not recognized as a risk factor present in the workplace, probably due to the lack of information and knowledge concerning changes on mycotoxin toxicokinetics when exposure occurs by inhalation and/or dermal absorption. These limitations might explain why there is no occupational exposure limit set for any mycotoxin [5]. Due to climate change, an increased magnitude and/or frequency of human exposure to mycotoxins is expected to occur, particularly in the current temperate regions of Europe. Additionally, mycotoxins that have not been commonly reported in food commodities from European Union (EU) countries might begin to occur as a result of changes in fungi species distribution associated with climate change [11,12,13,14]. For example, a wider dissemination of *Fusarium* fungi and its respective toxins is expected and will increase human (and animal) exposure and the prevalence of related health outcomes [15].

In summary, a different exposure pattern to mycotoxins might occur in the near future, or has already started to occur in the EU population, due to the ongoing climate change scenario observed in Europe. Therefore, there is a need to assess the current EU population’s exposure to mycotoxins that will serve as a baseline to monitor and compare future exposure, especially of vulnerable population groups, such as children and pregnant women, to assess the associated risk of developing health outcomes.

### 1.2. Mycotoxins in the Context of the HBM4EU Initiative

Human biomonitoring (HBM) consists of the measurement of a chemical or its metabolites (corresponding to the biomarkers of exposure) in body fluids or tissues, such as blood, urine, milk and hair [16,17]. HBM allows the determination of the internal exposure of individuals to chemicals, integrating all the exposure sources (e.g., food and air) and routes simultaneously [17]. Several HBM initiatives, from local to international, have been developed during the last few years in order to support the assessment of risk resulting from the human exposure to chemicals with proper exposure data. In addition to exposure biomarkers, the use of effect biomarkers in HBM studies contributes to establishing a link between internal exposure and early subclinical alterations that are suggestive of long-term health effects. Effect biomarkers consist of measurable biochemical, physiological, and behavioral effects within an organism that can be recognized as associated with an established or possible health impairment or disease [18].

The EU’s Environment and Health Action Plan 2004–2010 recognized the value of HBM and called for a coherent approach to biomonitoring in Europe [19]. Following two previous projects on HBM (COPHES and DEMOCOPHES), the European HBM Initiative (HBM4EU, 2017–2022) was set up with the main goal of coordinating and advancing HBM in Europe to provide science-based evidence for chemical policy development and improve chemical management [20].

A prioritization strategy to identify the chemicals or groups of chemicals to be studied under the HBM4EU [21] was implemented to determine and meet the most important needs of both policy makers and risk assessors, as well as the common national needs of participating countries and a broad range of stakeholders [22]. Following this strategy, mycotoxins were considered a priority substance group. Given the wide variety of compounds comprising mycotoxins, the EU Policy Board, the European Food Safety Authority (EFSA) and the Directorate-General for Health and Food Safety (DG SANTE) were consulted to allow the selection of the priority mycotoxins to be studied under the Project. The focus was set on deoxynivalenol (DON) and fumonisin B_1_ (FB_1_), for which data gaps regarding exposure and/or hazard are urged to be filled [23]. Although it has been recognized that exposure to mycotoxin mixtures is common and may lead to adverse health effects difficult to predict from the effects of single toxins, this topic was not addressed in this study. A scoping document containing a review of hazardous properties, exposure characteristics, policy relevance, technical aspects and the societal concern of mycotoxins, as well as a list of detailed policy-related questions, summarised in Table 1, identifying knowledge gaps and proposing research activities to close those gaps, was elaborated [24].

The policy questions addressed for these mycotoxins under the HBM4EU Initiative were driven by three main areas that constitute the steps of the chemicals risk assessment (RA), as illustrated at Figure 1: hazard assessment (including hazard identification and characterization), exposure assessment and risk characterization.

The exposure assessment step considers the choice of exposure biomarkers, respective specimens and the analytical methods for the evaluation of the magnitude and frequency of exposure, as well as the identification of the population groups that might be potentially highly exposed. The hazard assessment aimed at estimating internal exposures, identifying the main health effects reported in human and animal studies and contributing to elucidate the metabolic pathways and obtained toxicokinetics data that will support the choice of the most adequate exposure biomarkers. Finally, the adverse outcome pathway (AOP) framework [25] was used to organize the available knowledge from in vivo and in vitro studies to understand the biological mechanisms leading from the molecular or cellular perturbations to the health endpoints that are of regulatory relevance. A Human Biomonitoring Guidance Value (HBM-GV) was further developed, allowing the risk characterization of the target mycotoxins from the HBM data obtained for the European adult population.

### 1.3. Prioritized Mycotoxins: Occurrence, Toxicological Properties and Exposure Thresholds

Both DON and FB_1_ are produced by *Fusarium* species and occur predominantly in cereal grains. They are regulated in various cereal and derived products, being extensively measured in food in the EU through the monitoring programs in place [6].

DON is mainly found in wheat, oats, barley, maize and consequently also in cereal products such as breakfast cereals, bread, pasta and beer. It has been considered as immunotoxic, reprotoxic and a potential endocrine disruptor, and also induces intestinal disorders [26,27]. There is no evidence that DON could be carcinogenic to humans and, therefore, it is classified into Group 3 by the International Agency for Research on Cancer (IARC) [28]. A group tolerable daily intake (TDI) of 1 µg/kg bw/day was set for this mycotoxin and its derivatives, including 3-acetyl-deoxynivalenol (3-ADON), 15-acetyldeoxynivalenol (15-ADON), and deoxynivalenol-3-glucoside (DON-3G), based on reduced body weight gain in mice and assuming a chronic exposure to DON. However, an acute reference dose was also derived of 8 µg/kg bw per eating occasion, related to gastrointestinal effects that were reported in humans following exposure to high levels of DON in China [26]. The estimated mean chronic dietary exposure has been found to lie above the group-TDI in infants, toddlers and other children, and in highly exposed adolescents and adults, indicating a potential health concern [26].

Fumonisins (FB_1_, FB_2_, FB_3_ and FB_4_) are present mostly in maize and sorghum, with FB_1_ typically being the dominant one [29]. It is a suspected mutagen and a possible carcinogen, and thus classified by IARC in Group 2B [30]. It is teratogenic and associated with neural tube defects in the embryo. A group TDI value of 1 µg/kg bw/day was set for FB_1,_ FB_2_, FB_3_ and FB_4_ considering a BMDL_10_ of 0.1 mg/kg bw/day for megalocytic hepatocytes in mice [29]. There are no indications of acute adverse effects upon exposure to FB_1_.

Based on the available scientific knowledge on these two prioritized mycotoxins, the present article intends to gather the main achievements obtained under the HBM4EU project while responding to the initially raised policy questions. Based on the developed work, gaps have been identified and research needs are discussed for future human biomonitoring initiatives on mycotoxins.

## 2. Results Obtained for DON and FB_1_ in the Scope of the HBM4EU Initiative

### 2.1. Hazard Assessment

#### 2.1.1. Toxicokinetics (TK)

In the last few years, an increasing number of studies have been assessing the human exposure to mycotoxins in different countries using exposure biomarkers. DON, in particular, has been studied frequently because it is mainly excreted in urine: the parent compound and metabolites account for approximately 70% of the total dietary intake [31,32,33,34,35].

During the HBM4EU project, a literature search was performed on the TK of DON and FB_1–4_. Additional searches from other sources than scientific journals (e.g., EFSA reports and opinions) were also performed. Although excretion of DON and its main metabolites in humans has received ample attention in HBM studies, the TK in humans after single or multiple dosing has not been studied frequently. Vidal et al. carried out a human intervention study to unravel the urinary excretion profile and metabolism of DON and its modified form DON-3G [34]. Twenty volunteers were restricted to consuming cereals and cereal-based foods for 4 days. At day 3, a single bolus of 1 μg/kg body weight of DON and a single bolus of 1 μg/kg body weight of DON-3G, after a washing-out period of two months, was administered, and 24 h urine collection was performed. The urine was analysed for DON, DON-3G, 3-ADON, 15-ADON, deepoxy-deoxynivalenol (DOM-1), deoxynivalenol-3-glucuronide (DON-3-GlcA) and deoxynivalenol-15-glucuronide (DON-15-GlcA). The urinary biomarker analysis revealed that DON and DON-3G were rapidly absorbed, distributed, metabolized and excreted. Sixty-four percent of the administered DON and fifty-eight percent of DON-3G was recovered in the urine collected within 24 h. DON-15-GlcA was the most prominent urinary biomarker, followed by free DON and DON-3-GlcA.

In a follow-up manuscript [35], the authors developed biokinetic models for DON and DON-3G to determine: (1) the preferred (set of) urinary biomarker(s), (2) the preferred urinary collection period, and (3) a method to estimate the dietary exposure to these mycotoxins. The biokinetic models were based on three physiological compartments (gastrointestinal tract, liver and kidneys) and a known dietary exposure to these mycotoxins (i.e., the single DON or DON-3G bolus). This was used to estimate a reversed dosimetry factor (RDF). The main metabolic pathway for DON elimination is via glucuronidation, with DON-15-GlcA being the major metabolite [34,35,36,37,38]. DON and DON-3G are excreted relatively rapidly: within 12 h, 95% of the total elimination products are excreted via urine.

Van den Brand et al. [39] studied the excretion of DON after multiple daily dosing. These authors have assessed the relation between dietary DON intake and the excretion of its major metabolite DON-15-GlcA through time, in an everyday situation. For 49 volunteers from the EuroMix biomonitoring study, the intake of DON from each meal was calculated and the excretion of DON and its metabolites were analysed for each urine void collected separately throughout a 24 h period. The relation between DON and DON-15-GlcA was analysed with a statistical model to assess the residence time and the excreted fraction of ingested DON as DON-15-GlcA (f_abs_excr_). The estimated time in which 97.5% of the ingested DON was excreted as DON-15-GlcA was 12.1 h and the elimination half-life was 4.0 h. Based on the estimated f_abs_excr_, the mean RDF of DON-15-GlcA was 2.3. This RDF is comparable to the RDF reported in Mengelers et al. [35], and these can be used to calculate the amount of total DON intake in an everyday situation, based on the excreted amount of DON-15-GlcA. It was also shown that urine samples collected over 24 h was the optimal design to study DON exposure using HBM. So far, no physiologically based toxicokinetic models have been developed for DON.

Little is known on the renal excretion of FB_1–4_ in humans, presumably because animal studies have shown that FB_1_ is poorly absorbed from the gastrointestinal tract, rapidly cleared from the blood by the biliary route and preferentially excreted with the faeces. It is generally assumed that the metabolism and excretion of FB_2_, FB_3_ and FB_4_ are similar to that of FB_1_. The few data on the excretion of FB_1_ in humans consuming fumonisin-contaminated maize have suggested that the TK of FB_1_ in humans is similar to other mammalian species.

No TK models have been developed for FB_1–4_ in humans. Limited information is available regarding the TK of fumonisins in animals and it is mainly related with FB_1_. Previous studies have concluded that FB_1_ is poorly absorbed after oral ingestion in farm animals (e.g., swine, cow, laying hen) and experimental animals (rat, mouse, monkey) [29]. The bioavailable amount (less than 4% of the dose) is rapidly distributed to all organs and eliminated by biliary excretion without biotransformation. Faecal excretion vastly predominates over urinary excretion. Small amounts of partly hydrolysed and fully hydrolysed FB_1_ were detected as metabolites in faeces and are believed to be generated by the colonic microbiome. The EFSA reviewed in its scientific opinion the available data regarding the TK of FB_1_ [29,40]. According to the EFSA’s opinion, the vast majority of studies on fumonisins have been conducted with FB_1_ or with a natural mixture of fumonisins obtained from fungal cultures, which contained predominantly FB_1_ and smaller amounts of FB_2_ and FB_3_. No studies have been identified on the TK of FB_3_ and FB_4_, and only limited data have been identified on the modified forms HFB_1_ (hydrolysed FB_1_), pHFB_1_ (partially hydrolysed FB_1_) and NDF-FB_1_ (N-(1-deoxy-D-fructos-1-yl)-FB_1_) and no data on NCM-FB_1_ (N-(carboxymethyl)-FB_1_), although the latter compound is relevant as it was also detected in food samples. In general, the available studies considered that relating the FB_1_ concentration in urine to the dietary intake of FB of individual subjects is complicated due to interindividual variability and the rapidity of its clearance [41].

#### 2.1.2. Main Health Effects Identified for DON and FB_1_

Exposure to DON or FB_1_ has been shown to cause various adverse effects in in vitro and in vivo studies.

DON is suspected to be toxic for reproduction and it is able to cross the human placenta [42]. DON exposure affects H295R cell viability, steroidogenesis and gene expression indicating their potential as endocrine disruptors [27]. DON (and other trichothecenes) is immunotoxic, acting as a potent inhibitor of protein synthesis, stimulating the pro-inflammatory response and leading to oxidative stress generation [43].

FB_1_ acts by inhibiting ceramide synthases (CerS), key enzymes in sphingolipid metabolism. Besides being possibly carcinogenic to humans [30], in vivo studies have shown that the repeated exposure to this toxin causes liver and kidney toxicity [29] and may lead to liver and kidney tumorigenesis [30]. FB_1_ is clastogenic to mammalian cells [29] and also acts as a tumor promoter. Based on the results of animal studies, FB_1_ has been considered as a potential immunotoxic substance [44].

There are, however, no epidemiological studies that can provide unequivocal evidence of the chronic effects of these mycotoxins in humans. Only the acute effect upon DON exposure is well established in humans. With respect to FB_1_, one epidemiological study suggested an association between maternal FB_1_ exposure and neural tube defect (NTD) in the fetus [45]. In addition, Marasas et al. noted that several studies reported high occurrence of FB_1_ in certain areas where high frequencies of NTD in newborns were also observed [46]. This can be considered as circumstantial evidence for an association between FB_1_ exposure and NTD. Additionally, various animal studies support a link between FB_1_ and NTD [47,48,49,50,51,52,53,54].

Since a solid association between FB_1_ exposure and NTD could not be established in humans, the AOP framework was used within the HBM4EU to collect and organize in vitro and in vivo studies to support the circumstantial evidence on possible adverse effects of FB_1_ in humans [55,56].

#### 2.1.3. Development of AOPs for FB_1_ Based on Mechanistic Knowledge

An AOP describes the main events leading from a perturbation at the molecular level (the molecular initiating event, MIE) to an adverse effect on the organism or population (the adverse outcome, AO). The AOP framework helps to organize the data and evaluate the evidence that an event causes the next one. This curated mechanistic knowledge is accessible to users (e.g., risk assessors) through the AOP knowledge base (the AOP-KB, https://aopkb.oecd.org/index.html, accessed on 22 July 2022) and wiki (the AOP Wiki, https://aopwiki.org/, accessed on 22 July 2022). The AOP describing the biological mechanism that may underlie FB_1_-induced NTD in the developing embryo [55] (ID 449 in the AOP Wiki) was based on the mode of action described in the EFSA’s scientific opinion [29] and by Marasas et al. [46].

It proposes that FB_1_ triggers its effects (molecular initiating event, MIE) by inhibiting CerS, a key enzyme in sphingolipid metabolism (Figure 2). Indeed, FB_1_ is a structural analog of sphinganine (Sa) and sphingosine (So), the substrates of CerS, and a well-known inhibitor of CerS [57,58]. The drafted AOP describes two possible chains of events leading from CerS inhibition to NTD. The first route involves a decrease in folate uptake, which is known to be associated with the frequency of NTD. CerS inhibition may impact folate uptake through a decrease in complex sphingolipids (e.g., gangliosides), which are important constituents of membrane microdomains to which the folate receptor is presumably anchored [49,59,60,61]. The second route involves the inhibition of histone deacetylase (HDAC), which may induce NTD, as depicted in an existing AOP (under development) (ID 275 in the AOP-wiki, https://aopwiki.org/aops/275, accessed on 22 July 2022). In this scenario, HDAC activity is inhibited by phosphorylated forms of Sa and/or So, which are expected to increase in response to CerS inhibition [48,62,63].

Therefore, by describing biological mechanisms for FB_1_-induced NTD through inhibition of CerS, the proposed AOP reinforces the observations from limited human studies and provides a rationale for a causal relationship between exposure and health. It also highlights some uncertainties and gaps in the existing knowledge. For instance, it remains to be established how the first steps (key events) lead to a decrease in folate uptake, and the applicability of the chain of events in humans, given that the mechanism is largely based on in vitro and animal studies. Finally, it will be highly important to define the threshold levels of FB_1_ needed to trigger the MIE and compare them with predicted internal FB_1_ concentration.

Based on the mode of action proposed in the most recent EFSA scientific report, a putative AOP can be drafted for the DON-induced reduction of body weight gain [26,55]. The proposed MIE is the binding to ribosomes that would activate mitogen-activated protein kinases (MAPK), leading to the effects on body weight through two possible mechanisms. The first involves an inflammatory response in the intestine while the second involves the secretion of gut satiety hormones. These mechanisms find solid support from several in vitro and animal studies, but evidence for this association between DON exposure and reduced body weight gain in human is lacking.

#### 2.1.4. Effect Biomarkers in HBM Studies

In the scope of the HBM4EU Project, a literature search was carried out to find the most frequently used effect biomarkers in epidemiological studies addressing DON or FB_1_ exposure through HBM [64,65]. Even though the number of HBM studies focused on mycotoxin exposure has been increasing in the last decade, only very few of them have included the analysis of effect biomarkers. Several studies have suggested that the increase of the Sa/So ratio in biological fluids can be used as a sensitive biomarker of fumonisin exposure and early biological effects [41,66,67]. The validity of the Sa/So ratio as a biomarker in humans remains, however, uncertain [67,68]. This is partly because Sa and So occur and vary naturally in human blood [68,69]. Furthermore, there is no human reference value for physiologically normal levels of these sphingoid bases or the Sa/So ratio. In this context, the proposed AOP provides some mechanistic support for the Sa/So ratio as a biomarker of an early effect. However, although the increase in Sa/So ratio is an expected consequence of the MIE, it is not a key event of the AOP per se. Based on the AOP that was developed for FB_1_, other effect biomarkers may be proposed. As FB_1_ is recognized as an inhibitor of HDAC, a central regulator of gene expression, a more open chromatin structure, resulting in deregulated gene expression is expected following exposure to this mycotoxin. Therefore, in vitro assays that target chromatin structure or downstream gene expression can be used to start the development of novel effect biomarkers.

Concerning DON exposure, no effect biomarkers were found that could be associated with its adverse health effects.

#### 2.1.5. Derivation of a HBM-GV for DON

Within the HBM4EU, a HBM-GV of 23 μg total DON/L (95% CI 5–33 μg total DON/L) was derived for total DON concentrations in 24 h urine samples [70]. Depending on the analytical approach taken to analyse the urine, an HBM-GV of 20 μg DON-15-GlcA/L (95% CI 7–39 μg DON-15-GlcA/L) can also be applied for this main (phase II) metabolite of DON [70]. As more than 90% of DON is excreted approximately 12 h after ingestion, almost all ingested DON is captured in a 24 h urine sample (provided that it includes the first morning urine sample at the end of the collection period) [35,39].

The HBM-GV for DON was derived for 24 h urine samples and should not be applied to morning urine or other spot urine samples, as the elimination half-life of DON is too short (approximately 3 to 4 h). The variation in spot urine samples will consequently be too large (larger than the dosing interval calculated for a 24 h urine sample). This, of course, has practical consequences, as many studies (including the HBM4EU Aligned Studies on DON) collected spot urine samples rather than 24 h urine samples [71,72]. Although there are studies that report a good correlation between morning urine and 24 h urine samples [73], a fraction to convert a morning urine sample to a 24 h urine sample cannot be used. The uncertainty around this fraction would be too large considering the short excretion half-life of DON, the variation of DON intake throughout the day and the lack of information regarding the urine voids/discharges of the volunteers between their last meal and first morning urine collection. In addition, the HBM-GV was derived by using an average, body-weight-adjusted, urinary flow rate based on the 24 h urine volumes of the 20 volunteers in a human intervention study [35]. This flow rate can differ between populations, which will affect the derived HBM-GV. Considering that the study population of Mengelers et al. [35] was small, the average flow rate may be prone to fluctuations. This is a general observation that needs to be kept in mind when deriving a HBM-GV from a HBGV that was based on an animal study, only using ‘external’ doses.

#### 2.1.6. Responses to Policy Questions on Hazard Assessment

Concerning the policy question defined under the HBM4EU, “Are there toxicokinetics data for the target mycotoxins and which are their limitations?”, the answer is affirmative for DON, since a dedicated model was developed under the HBM4EU, but not for FB_1_. Future efforts should be made to increase the knowledge on FB_1_ toxicokinetics and, consequently, contribute to a better human risk assessment. Currently, the toxicokinetics of DON after oral intake appear to be better characterized. Nevertheless, occupational exposure to DON can occur through inhalation of, e.g., contaminated flour dust, and the absorption characteristics for this route of exposure are still unknown. Physiologically based toxicokinetic models, including absorption by inhalation and oral absorption could contribute to the risk assessment of occupational exposure.

In response to the question “Which are the key events that determine the chronic health effects of the target mycotoxins?”, an AOP was provided, in which several key events might determine FB_1_-induced neural tube defects. However, data on long-term effects from low-dose continuous exposure are lacking and human studies are insufficient. Moreover, associations between key events/potential effect biomarkers and the health effects are not established. In fact, there are difficulties in reliably establishing exposure to FB_1_ in humans, since some key events are technically challenging, and causality for chronic effects is difficult to establish with human data.

Concerning the question “Which are the most frequent AOP-based effect biomarkers for the prioritized mycotoxins?” and after a literature review, the most specific effect biomarker identified was related to the inhibition of CerS: Sa and So levels in blood or urine. However, there are insufficient data available on the effect biomarkers for FB_1_ and no studies for DON. For example, omics-based methods, such as transcriptomics, can be used to identify a number of differentially expressed genes involved in previously identified KE that can be candidate effect biomarkers. The value of those candidates can start to be evaluated through in vitro functional assays and their sensitivity and reliability can then be assessed in epidemiological studies.

Finally, concerning the question “Is it possible to set a HBM-GV for the target mycotoxins”: indeed, a HBM-GV was derived for DON, but it is only applicable for 24 h urine samples, which increases the uncertainty associated with its use for spot urine samples.

### 2.2. Exposure Assessment

#### 2.2.1. Mycotoxin Exposure Biomarkers

Exposure to DON includes not only the ingestion of DON as such, but also its derivatives 3-ADON and 15-ADON, and the plant metabolite DON-3G, with a relative contribution in food of 10%, 15% and 20%, respectively [26]. Upon uptake, these three derivatives are biotransformed into DON [34,74]. The free DON is then glucuronidated and excreted in urine as DON-15-GlcA, free DON and DON-3-GlcA. After enzymatic deconjugation, the glucuronides are converted into DON; DOM-1 and DON-3-sulfate are minor human metabolites [75]. DON is well absorbed, rapidly cleared from the blood and mainly completely excreted through urine in 6–16 h. This makes urine the target matrix for human biomonitoring of DON. In HBM, urine sampling is mostly performed by collecting spot urine samples, often first morning urine, which is suboptimal for assessment of daily exposure (see also Section 2.1.1). In urine, DON is mostly present as DON-15- GlcA, DON-3- GlcA, and free DON (fDON) (ratio 58:27:14) [34]. There has been some debate on the preferred biomarker of exposure: either DON-15-glucuronide or total DON (tDON, sum of fDON and its glucoronides) (after deconjugation). Incomplete deconjugation was raised as a concern by [76] but was contradicted by [75]. As long as the appropriate enzyme and conditions are used, complete deconjugation is obtained and therefore tDON is an adequate biomarker of exposure. From an analysis perspective, the measurement of tDON is advantageous because the analytical reference standards of this compound and its isotope-labelled analogue are readily available, in contrast to DON-15-GlcA. Within the Quality Assurance/Quality Control (QA/QC) program of the HBM4EU, an interlaboratory comparison investigation (ICI) was performed amongst six laboratories (one withdrew during the program). In a period of six months, three sets of two different urine samples of humans exposed to DON (/derivatives) had to be analysed for tDON, using methods capable of quantifying DON at 0.5 ng/mL or lower. Concentrations in the quality control urines ranged from 0.5–33 ng/mL. From this exercise, four laboratories generating comparable results were identified (relative standard deviations of concentrations found were around 13% in most cases, 27% for one urine sample). These laboratories performed the analysis of samples from the HBM4EU Aligned Studies.The volume of urine needed for analysis varied from 0.5–3.0 mL. Laboratories were free to use their own in-house method. All four laboratories used the same enzyme (*E. coli* based ß-Glucuronidase) for deconjugation (15–24 h, pH 7, 37 °C). Extraction/cleanup was performed either by using immuno-affinity columns or with more generic solid phase extraction. DON was measured using LC-MS-MS (liquid chromatography with tandem spectrometry) with isotope-labelled DON as the internal standard. More details can be found in [75].

Compared to DON, the biomonitoring of fumonisins is less straightforward. The details have been described in [37,75] and are briefly summarised here. Fumonisins appear to be poorly bioavailable and mainly excreted through faeces. Fumonisins are stable and the parent compounds are considered the target biomarker of exposure, with a focus on FB_1_, since this is the most abundant fumonisin in contaminated food. Despite the fact that only a fraction (<4%) is excreted through urine, urine is the most common matrix used for biomonitoring of fumonisins. Blood has not been widely reported for the biomonitoring of fumonisins. Hair and breast milk have been used as human matrices, both less suited for exposure assessment purposes. In urine, LOQs (limits of quantification) in the pg/mL are required to avoid left-censored data in the general population in Europe. Typical methods involve an extraction/cleanup step using immuno-affinity columns or solid phase extraction and LC-MS-MS measurement. In principle, the determination of DON and FB_1_, and other mycotoxins, can be combined. However, this usually results in compromises in terms of LOQ.

#### 2.2.2. Exposure Scenarios and Levels of the EU Population

In order to assess and characterize the exposure of the European population to mycotoxins, two approaches were used, i.e., collection of exposure data from a literature search and use of the HBM data obtained in the HBM4EU Aligned Studies [77].

##### Literature Search

The literature search was performed with the objective of identifying publications on mycotoxin HBM in Europe from the last two decades (2000–2022). It allowed us to obtain aggregated data from all the published studies (papers, reviews and other reliable sources). The authors/data owners were contacted and invited to provide individual data through a data transfer agreement within the scope of the HBM4EU Project. Both data sets (aggregated and individual) were used for statistical assessment of European population exposure to mycotoxins in order to characterize exposure determinants and geographical variability for the prioritized mycotoxins, including DON (tDON) and FB_1._ If the studies reported mycotoxins other than DON and FB_1_, that information was also extracted and compiled in the database.

A literature search using the terms “exposure”, “biomonitoring” and “mycotoxins” was performed using the databases of PubMed, Scopus and Web of Science. HBM data concerning mycotoxins and their metabolites (number of participants/samples studied, type of biological samples collected, mycotoxin biomarker concentration, methods and analytical conditions, geographical area, sampling year and available relevant demographic information such as age, sex, and area of residence) were systemized in a single database. The literature search identified 106 publications on human biomonitoring, including approximately 78 mycotoxins and metabolites. The detailed information directly retrieved from the publications was stored in the above mentioned database.

From the publications identified for DON, four individual data collections from four European countries (Portugal, Italy, Norway and the United Kingdom) were shared under the project and the respective metadata files included at the Information Platform for chemical monitoring (IPCHEM, https://ipchem.jrc.ec.europa.eu/, accessed on 27 July 2022). For aggregated data, studies reporting mean and/or median levels, unadjusted (ng/mL) and creatinine-adjusted (ng/mg crt), on morning/spot or 24 h urine samples and covering exposure determinants such as population group, sex and geographical regions were identified.

The results obtained from this literature search showed that exposure to tDON in the European population was generalized, affecting different age groups of the population (children, adolescents, adults, the elderly), both sexes and all regions (Northern, Southern, Western, Eastern). Of the age groups mentioned, adults (70% of the studies) were the most evaluated in terms of exposure to tDON, followed by children (12%), adolescents and the elderly (9% each). Two population groups were also identified—pregnant women and vegetarians—with five and three studies, respectively. With regard to sex, a similar number of studies was found for both sexes together (MF-38%) and separately (M-30% and F-32%). Concerning geographic coverage, 40% of the studies for tDON were carried out in Northern Europe, with Southern and Western Europe showing values of 30 and 27%, respectively, and only one study (3%) covering the Eastern region.

Due to the heterogeneity of published studies with aggregated data (missing types of samples, population samples with both sexes and with a variable number of individuals), their comparison and the analysis of exposure determinants is extremely difficult. To make this possible, it is necessary to further harmonise studies in Europe in order to be possible to define policy actions.

##### Aligned Studies

The HBM4EU Aligned Studies were studies targeting the general population that were aligned concerning the sampling period (2014–2020), age group (20–39 years for the adult studies), sampling size (≤300 participants per study), biomarkers analysed and data collected. Studies performed in hotspots, in patients or in occupational groups were excluded. As the aim of the aligned studies was to characterize exposure at the European level, countries from the four geographical regions of Europe according to the United Nations geo-scheme (North, East, South, West) were included. Concerning mycotoxins, tDON was selected as a biomarker to characterize exposure to DON, and six countries contributed to data, namely France, Germany, Iceland, Luxembourg, Poland and Portugal. However, since data from Iceland was not quality-assured by the HBM4EU QA/QC program, it was excluded from the calculations of the European exposure values to tDON.

Results from the Aligned Studies, comprising 1099 individuals from five European countries, have shown that the European population is exposed to DON with a geometric mean of 5.59 µg/L (5.09 µg/g crt) and a 95th percentile of 36.15 µg/L (31.21 µg/g crt). Concerning the geographic variability, differences were observed between regions, with Western Europe presenting the lowest exposure value and Eastern Europe the highest. Concerning socio-demographic factors, no difference was observed for sex, but differences were observed concerning the educational level and the degree of urbanization of the residence of the individual. Individuals with a low educational level had higher exposure levels than individuals with a higher educational level; individuals living in rural areas had higher exposure levels than individuals living in cities and in towns/suburbs. Concerning the sampling season, samples collected in the winter presented the lowest levels (S. Namorado, personal communication; paper in prepation).

These studies have allowed us to compare the results of exposure to tDON from different European countries, but also demonstrated the need for a more extensive evaluation, considering the higher exposure levels observed for Poland and the lack of mycotoxins exposure data for countries from Eastern Europe. The results also showed the need for an adequate study design with sample collection distributed throughout the year to take into account season variability. The differences observed concerning educational level and residence area showed that specific groups of the population have higher exposure and should be targeted in information campaigns aimed at informing the general public to reduce exposure.

#### 2.2.3. Responses to Policy Questions on Exposure Assessment

Concerning the question: “Are there validated and harmonised analytical methods to assess the target mycotoxins exposure?”, the answer is affirmative, since a set of laboratories is now available that can generate comparable HBM data for urinary tDON. However, no reference materials are available for DON and FB_1_ in urine, nor an analytical reference standard for DON-15-GlcA and its isotope analogue. Currently, no continuation of a proficiency testing program for HBM DON and FB_1_ biomarkers in urine is possible. Other questions were raised concerning exposure, namely “What are the current exposure levels of the European population to DON and FB_1_? Does the exposure to mycotoxins differ among different population groups? Which are the main exposure determinants?”. These questions were partly answered by the results from the Aligned Studies and bibliographic search. In the scope of the HBM4EU Aligned Studies, it was possible to characterize the current exposure level of the European population to DON and to evaluate the main factors associated with this exposure. It was also possible to evaluate to some degree the geographic variability, as data from several European regions was available; however, considering the low number of studies from Eastern and Southern Europe, the results should be considered with caution and further studies including more countries should be developed. Future studies will also allow the evaluation of time trends.

The results obtained from a complimentary literature search spanning the last twenty years confirmed that exposure to tDON in the European population is generalized. However, due to the heterogeneity of the published aggregated data, no comparative study or mean exposure value was derived. Instead, individual data from some European countries have been shared under an Agreement Protocol and will be analysed for exposure determinants and geographical variability. No trend analysis was performed due to the lack of individual data in the same periods.

No data was obtained for urinary FB_1_, since there is no ideal biomarker of exposure due to very low urinary excretion.

### 2.3. Risk Characterization

#### 2.3.1. Approaches to the Risk Characterization of DON

Risk characterization was first performed using data from the studies identified in the literature search performed [77].

The results (Figure 3) show that children and pregnant women, groups traditionally considered vulnerable, presented the highest risk. The exposure of workers to DON was also considered in this assessment, and although a potential health risk was not identified, it is important to emphasize that the referred studies reported a statistically significant difference between workers and control groups, confirming that the occupational environment can have an important role in the exposure to DON.

Risk characterization using results from the HBM4EU Aligned Studies followed two different approaches: (i) the estimation of probable daily intake calculated through reverse dosimetry from HBM data and comparison with the group TDI for DON (1 µg/kg bw/day), and (ii) the direct comparison of HBM results from the aligned studies with the HBM-GV (23 μg total DON/L, 95% CI 5–33 μg total DON/L) [70].

The results from the studies conducted in the adult populations of Poland, France, Portugal, and Luxembourg showed that the highest percentiles of exposure (P90 and P95) represented a potential health concern for the most exposed individuals, since the hazard quotient (HQ) determined was above one. However, the mean and median levels of exposure were considered as not representing a concern for health. The results obtained from the aligned studies conducted in Iceland and Germany revealed that exposure to DON did not represent a health concern, since the HQ was below one for all percentiles of exposure.

Concerning exposure to FB_1_, due to a high uncertainty associated with estimates, it was not considered adequate to develop a risk characterization. Additionally, risk characterization using the data from the literature search was not performed due to lack of access to data at the individual level and the consequent high uncertainty in estimates.

#### 2.3.2. Responses to Policy Questions on Risk Characterization

Concerning the policy question defined under the HBM4EU “Is the risk associated to human exposure to these mycotoxins characterised?”, the answer is affirmative, with the obtained results confirming that the European population is exposed to DON and that a fraction of this population is, to some extent, exposed to levels that might represent a potential health concern. The inclusion of mycotoxins’ HBM data in risk assessment is important since it represents the internal exposure dose from all sources and by all routes of exposure at the individual level, thus reducing the uncertainties associated with risk assessments performed at the population level and/or indirect approaches (e.g., through a combination of occurrence in food and food consumption data) [17].

However, some weaknesses in this assessment should also be discussed. The use of HBM data for mycotoxins implies an extensive knowledge of metabolism, but there are still some gaps regarding mycotoxins’ toxicokinetic data that may hamper a proper risk assessment [78]. In the context of risk assessment for regulatory purposes, it is important to consider all these aspects. Regarding compounds for which a reduced knowledge on metabolism is available, the issue of uncertainty in estimates remains and the limitations of the HBM-GV should be described in detail.

Another issue is the relevance of considering and collecting information on all the possible exposure scenarios/sources that might explain the HBM results (e.g., diet, workplace environment). It should be noted that collecting contextual data to identify the exposure scenarios/sources facilitates target regulatory and policy actions aiming to prevent exposure.

## 3. Overview of the Main Achievements of the HBM4EU on Mycotoxins and Future Trends

### 3.1. Policy Relevance

Table 2 summarises the policy relevance of the main achievements on prioritized mycotoxins obtained under the HBM4EU, while responding to the initially raised policy questions. The main achievements related to DON concerned increased knowledge on the European population’s exposure and risk characterization, whereas for FB_1_ they were mainly focused on its hazards, allowing the establishment of an AOP for neural tube defects. These achievements are expected to increase trust and the use of HBM data for regulatory purposes, i.e., for assessment of risks from exposure to the studied mycotoxins.

### 3.2. Future Trends and Research Needs

Despite the great progress made, several aspects need further research. Regarding the analytical methods, there is still a need for interlaboratory comparison programs for exposure biomarkers other than DON, as well as certified reference materials, to promote comparability among research groups. The work performed under the HBM4EU revealed an opportunity to improve multi-mycotoxin biomarker methods and to more efficiently and cost-effectively generate HBM mycotoxin data, as well as proficiency testing of multi-mycotoxin methods. Collection of 24 h urine samples for DON are encouraged in order to decrease risk assessment uncertainties related to the collection of spot urine samples. Regarding fumonisins, future efforts should be made to identify an alternative biomarker of exposure for FB_1_ as well as to increase the knowledge of their toxicokinetics after oral intake and, consequently, contribute to a better human risk assessment. Physiologically based toxicokinetic models have yet to be developed for DON.

The design of human studies to explore relationships between the studied mycotoxins exposure and identified health outcomes (e.g., neural tube defects and hepatotoxicity for FB_1_) is also needed. There is a need to validate Sa/So as biomarkers for FB_1_, better define the level of confidence in the AOPs and provide quantitative information, better characterize the proposed biomarkers of effects, and identify effect biomarkers for DON based on an AOP. The work developed under the HBM4EU on the hazard assessment of FB_1_ allows advances in some directions to develop biomarkers related to the mechanistic knowledge from in vitro and in vivo studies that was organized under AOPs. Indeed, effect biomarkers can assist in the assessment of exposure to single mycotoxins and, more importantly, to their mixtures, by contributing to the interpretation of exposure biomarker data and by bridging exposure to health outcomes.

Moreover, mycotoxins’ combined effects are also worth investigating given that they are likely to be additive or even synergistic. In fact, human exposure may be within the TDI for each of the mycotoxins, but the sum of both can exceed the level deemed to be safe and thereby give rise to health effects that should be considered for regulatory purposes.

The present study highlights the relevance of knowing the current exposure to DON in the EU and raises awareness on risks to human health associated with exposure to this mycotoxin. In future, similar studies must be conducted, comprising a broader spectrum of mycotoxins with health implications and widening EU geographical coverage. Other opportunities for future research highlighted during this initiative included the need for development of guidance for setting up biomonitoring campaigns to allow a proper comparison among studies, and building on the work developed under HMB4EU to adjust HBM-GV with results obtained through collection of first morning urine samples. The inclusion of mycotoxin HBM data in a risk assessment is important since it represents the internal exposure dose from all sources and by all routes of exposure at individual level, thus reducing the uncertainties associated with risk assessment performed at the population level and/or indirect approaches. The impact of climate change on population exposure to mycotoxins should also be monitored regularly in order to take actions to prevent exposure and to anticipate potential health issues.

## 4. Methods

This paper is the result of the work of the group of experts within the HBM4EU project dedicated to mycotoxins. This working group met and discussed with the purpose of gathering the information provided by the project and described in detail the outputs that have policy relevance at EU level. In addition, the working group also identified the topics that need further discussion and research actions considered in the present paper (Section 3.2).

The results are presented according to the three general steps of risk assessment. Within the hazard assessment, four different domains were included for the prioritized mycotoxins: toxicokinetics, main health effects and related effect biomarkers, development of an adverse outcome pathway (AOP) based on mechanistic knowledge and derivation of HBM-GV. Within the exposure assessment, three main areas were focused on: exposure biomarkers, exposure scenarios and exposure levels of the EU population, considering a literature search and the HBM4EU-aligned studies. Within the risk characterization step, the risk associated with the exposure levels characterized through the aligned studies were included. After each step, a SWOT analysis was conducted and strengths and weaknesses were identified.

## Figures and Tables

**Figure 1 toxins-14-00826-f001:**
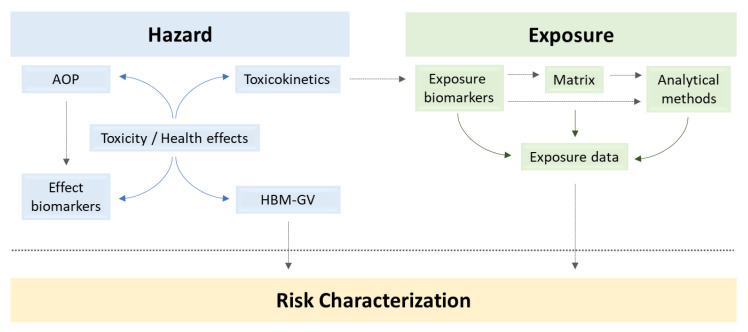
Human Biomonitoring and Risk Assessment of mycotoxins (compliance with RA components). AOP—adverse outcome pathway, HBM-GV—Human Biomonitoring Guidance Value.

**Figure 2 toxins-14-00826-f002:**
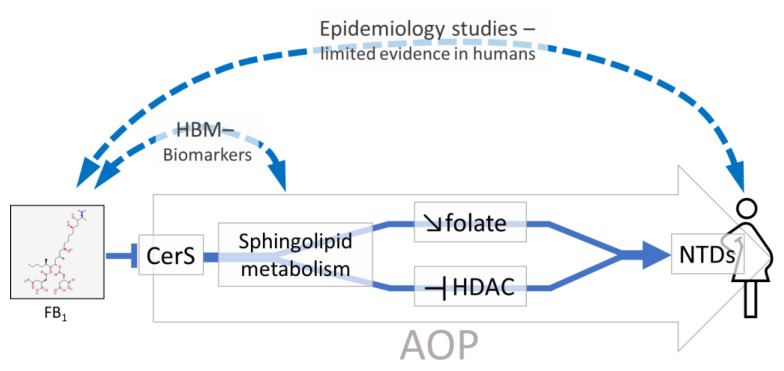
Simplified scheme of a proposed AOP depicting the mechanism for FB_1_-induced neural tube defects (NTDs).

**Figure 3 toxins-14-00826-f003:**
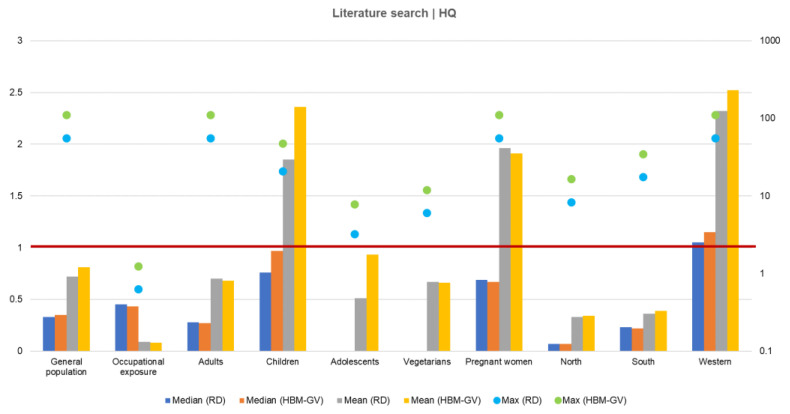
Risk characterization through hazard quotient calculation. *Y*-axis = median and mean hazard quotient (HQ); Secondary *Y*-axis = maximum HQ (logarithmic scale).

**Table 1 toxins-14-00826-t001:** Policy questions on mycotoxin human biomonitoring under the HBM4EU according to risk assessment steps.

	Policy Questions on Mycotoxins under HBM4EU
Hazard assessment	Are there toxicokinetics data for the target mycotoxins and what are their limitations? What are the key events that determine the chronic health effects of the target mycotoxins? What are the most frequent AOP-based effect biomarkers for the prioritized mycotoxins? Is it possible to set HBM guidance values for the target mycotoxins?
Exposure assessment	Are there validated and harmonised analytical methods to assess the target mycotoxins’ exposure? Which are the current exposure levels of the European population to DON and FB_1_? Does the exposure to the target mycotoxins differ among different population groups? Which are the main exposure determinants?
Risk characterization	Is the risk associated with human exposure to these mycotoxins characterized?

**Table 2 toxins-14-00826-t002:** Policy relevance of main achievements on DON and FB_1_ under the HBM4EU.

Policy Relevance of Main Achievements
Total DON exposure assessment of the European population through human biomonitoring (internal exposure) using a harmonised and HBM4EU-qualified method was, for the first time, attained.
The characterization of the risk from exposure to DON was improved, given the possibility of directly comparing the internal dose measured with the newly derived human biomonitoring guidance value, although some associated uncertainties must still be considered.
The assessment of DON exposure and the associated risk characterization contributed to close a data gap on the Eastern European population exposure and evidenced some concern regarding the most exposed population groups that deserve further action.
Comparison of dietary exposure through food occurrence and the internal dose of DON is now possible due to the development of a toxicokinetics model, although some refinement might still be needed to evolve to a physiologically based TK model.
The characterization of the internal exposure to DON can be used as a baseline level to monitor exposure of the European population to this mycotoxin, e.g., considering climate changes expected.
The use of the AOP framework to organize mechanistic data on FB_1_ provided biological plausibility to the development of congenital anomalies (neural tube defects) associated with early life exposure to FB_1_.
Effect biomarkers have been rarely analysed in epidemiological studies concerning DON and FB_1_ (and other mycotoxins) exposure and need to be further developed.

## Data Availability

Not applicable.

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
