# Peer review of "Current Advances, Research Needs and Gaps in Mycotoxins Biomonitoring under the HBM4EU—Lessons Learned and Future Trends"

_toxins, 2022, doi:10.3390/toxins14120826_

Round 1
Reviewer 1 Report
The present review entitled “Current advances, research needs and gaps in Mycotoxins Bio-2 monitoring under HBM4EU – lessons learned and future trends” sums several findings conducted under the European Human Biomonitoring Initiative (HBM4EU), specifically for mycotoxins deoxynivalenol and fumonisin B1.
These results are of great scientific significance for pushing forward the understanding of how mycotoxins impact public health, allowed by the great number of studies, samples and laboratories involved within the project reviewed by this manuscript. These results also set standards for future exposure assessments of DON and FB1, contributing to conduct more realistic approach in this type of studies.
The manuscript is well written and organized, and the approach towards policy give even more value to the presented results. Conclusions corresponding to each section of the review are supported by the presented findings, which also clearly responded to some of the main goals established by the reviewed project.
The fact that gaps and limitations of the conducted investigations have been also addressed help to contextualize the presented findings and understand which direction future studies must follow in order to cover these flaws. Besides, this review established a frame for future studies to also provide consistent results at several points of risk assessment studies. For all of the above, I believe this is a work suitable for this journal in its current form.
Author Response
The authors thank the reviewer for his fruitful and positive comments. They also acknowledge the positive references to manuscript organisation, interest and scientific importance.
Reviewer 2 Report
The paper is well organised, cited literature is relevant and the topic is of interest. In my opinion the methodology and approach to HBM4EU are well presented and clear and. The results could be useful to scientific community. I think that the paper can be accepted in the present form.
Author Response

(The authors gave the same response as above.)

Reviewer 3 Report
The manuscript presented the current advances attained under HBM4EU, research needs and gaps in in mycotoxins biomonitoring, and reviewed several policy questions hazard characterization, exposure and risk assessment. Some suggestions have been made in order to improve the quality of this paper.
1. Why choose the mycotoxins deoxynivalenol (DON) and fumonisin B1 (FB1) as priority substances to studied under HBM4EU. In fact, other mycotoxins such as patulin, OTA is also widely found in food and their derived products
2. In this manuscript, the single of DON and FB1 was considered and to study the hazard characterization, exposure and risk assessment. However, co-exposure to several mycotoxins has been commonly found. It is necessary to consider their combined effect on human health.
3. The boiling points of DON and FB1 are very high, and they are stable to heat, the mycotoxin is how to be inhaled or dermal absorbed by human in workplace.
4. The toxic effect of DON, 15-ADON, 3-ADON, DON-3G are very different, why they have the same tolerable daily intake (TDI) of 1 µg/kg. In addition, DON is how to transform to 15-ADON, 3-ADON, DON-3G, DOM-1, DON-3-GlcA, DON-3-GlcA.
5. Line 488-489,“differences were observed between regions, with Western Europe presenting the lowest exposure value and Eastern Europe the highest”,why have this difference between Western Europe and Eastern Europe?
Specific comments:
1. Line 61-63, please rewrite this sentence of “mycotoxins not yet commonly reported in the food commodities avail- 61 able in the European Union (EU) might begin to occur due to changes in fungi species 62 distribution in an attempt to adapt to climate change”.
2. Line 134, “classified in” change to “classified into”.
3. Line 149, “There were” change to “There are”.
4. For DON, DON-3G, 3-ADON, 15-ADON, the first appear, the full name is used, then for the latter appear, just use abbreviation.
5. Please add the reference for Part 2.2.2.
6. Line 240, please delete “is”.
7. Line 488, “diferences” change to “differences”.
8. Please check Line 411, “The details have been described in [80] and [37] and are briefly summarized here.”. the reference cited is right?
Author Response
The authors thank the reviewer for the detailed and comprehensive comments that improved the manuscript quality and helped clarifying some issues.
General and specific comments performed by reviewer 3 are answered point-by-point, as follows below.
- Why choose the mycotoxins deoxynivalenol (DON) and fumonisin B1 (FB1) as priority substances to studied under HBM4EU. In fact, other mycotoxins such as patulin, OTA is also widely found in food and their derived products
The authors thank and agree with the reviewer comment. In fact, patulin as well as other important mycotoxins with significant health impact (such as ochratoxin A, aflatoxins) could be selected as prioritised compounds. Although an explanation was given in lines 96-99, we modified them to improve the explanation, as follows (lines 97-102):
“Given the wide variety of compounds comprised in the group of mycotoxins, the EU Policy Board, the European Food Safety Authority (EFSA) and the Directorate-General for Health and Food Safety (DG SANTE) were consulted to allow the selection of the priority ones to be studied under the Project.The focus was set on deoxynivalenol (DON) and fumonisin B1 (FB1), for which data gaps regarding exposure and/or hazard urged to be filled [23].”
- In this manuscript, the single of DON and FB1 was considered and to study the hazard characterization, exposure and risk assessment. However, co-exposure to several mycotoxins has been commonly found. It is necessary to consider their combined effect on human health.
The authors thank and agree with the comment. In fact, this important aspect is highlighed in lines 51-53, where it is stated “Furthermore, co-exposure to several mycotoxins has been commonly found. However, their combined effect on human health needs to be considered and taken on board at regulatory level when defining maximum concentrations [10].”
It is also addressed under 3.2. Future trends and research needs, lines 621-624: “Moreover, mycotoxins’ combined effects are also worth to investigate given that they are likely to be additive or even synergistic. In fact, human exposure may be within the TDI for each of the mycotoxins but the sum of both can exceed the level deemed to be safe and thereby give rise to health effects that should be considered for regulatory purposes.”
Despite those references to mixtures in the text, we added the following sentence (lines 102 – 104) to enphasize its relevance: “Although it has been recognized that exposure to mycotoxins’ mixtures is common and may lead to adverse health effects difficult to predict from the effects of the single toxins, this topic was not addressed in this study.“
- The boiling points of DON and FB1 are very high, and they are stable to heat, the mycotoxin is how to be inhaled or dermal absorbed by human in workplace.
The authors thank the comment and try to clarify that occupational exposure to mycotoxins concerns other exposure routes (than digestion) that do not imply boiling temperature. According to Viegas et al (reference 5, lines 670-671 this manuscript) “Most mycotoxins are not volatile. However, mycotoxins can be present in airborne dust and in the fungal spores and fragments. Therefore, airborne dust, spores, and hyphae fragments can act as carriers of mycotoxins to the lungs and potentially, exposure in occupational settings occurs essentially via inhalation, particularly in the form of airborne dust”.
We added more information to the sentence in the introduction to give more complete information on this issue (lines 35-40): “ Food …..In addition, inhalation and dermal routes can also contribute to exposure in certain occupational settings, mainly during tasks involving high exposure to organic dust (e.g. storage work, loading, handling, or milling contaminated grains, waste and feed) because airborne dust, spores, and hyphae fragments can act as carriers of mycotoxins to the lungs [4,5].”
- The toxic effect of DON, 15-ADON, 3-ADON, DON-3G are very different, why they have the same tolerable daily intake (TDI) of 1 µg/kg. In addition, DON is how to transform to 15-ADON, 3-ADON, DON-3G, DOM-1, DON-3-GlcA, DON-3-GlcA.
The authors thank and try to clarify the reviewer comments.
Concerning the group TDI for DON, and according to EFSA (ref 26, lines 725-728 this manuscript) “DON is rapidly absorbed, distributed, and excreted. Since 3-Ac-DON and 15-Ac-DON are largely deacetylated and DON-3-glucoside cleaved in the intestines the same toxic effects as DON can be expected. The TDI of 1 µg/kg bw per day, that was established for DON based on reduced body weight gain in mice, was therefore used as a group-TDI for the sum of DON, 3-Ac-DON, 15-Ac-DON and DON-3-glucoside.” Hope this could help to clarify the question from the reviewer.
Regarding the transformation: 15ADON, 3-ADON and DON3G are fungal and plant metabolites of DON. DON-3-GlcA, DON-15-GlcA and DOM-1 are mammalian metabolites. Details can be found in the EFSA report.
The following sentence was added to the text (lines 388-390): “The free DON is then glucuronidatedand excreted in urine as DON-15GlcA, free DON and DON-3-//GlcA//. After enzymatic deconjugation, the glucuronides are converted into DON; DOM-1 and DON-3-sulfate are minor human metabolites [80].”
- Line 488-489,“differences were observed between regions, with Western Europe presenting the lowest exposure value and Eastern Europe the highest”,why have this difference between Western Europe and Eastern Europe?
Thank you for your question. A paper on results on the exposure levels in the HBM4EU Aligned Studies is currently under preparation and it will include the analysis of the exposure determinants for each of the countries that may help to explain this geographical difference in DON exposure. In the current paper we have decided to include the general results that will be detailed in the other paper, but considering the lack of pre-existing data for Eastern Europe countries we had included, in the present manuscript, the sentence “need of a more extensive evaluation considering the higher exposure levels observed for Poland”. We hope that this explanation will help to clarify the reviewer´s question.
Specific comments:
- Line 61-63, please rewrite this sentence of “mycotoxins not yet commonly reported in the food commodities avail- 61 able in the European Union (EU) might begin to occur due to changes in fungi species 62 distribution in an attempt to adapt to climate change”.
Corrected.
The sentence “mycotoxins not yet commonly reported in the food commodities available in the European Union (EU) might begin to occur due to changes in fungi species distribution in an attempt to adapt to climate change [11–14]” was replaced by “mycotoxins that have not been commonly detected in the food commodities from European Union (EU) countries might begin to occur as a result of changes in fungi species distribution associated with climate change [11–14]” (lines 61-63).
- Line 134, “classified in” change to “classified into”.
Corrected (line 133).
- Line 149, “There were” change to “There are”.
Corrected (line 154).
- For DON, DON-3G, 3-ADON, 15-ADON, the first appear, the full name is used, then for the latter appear, just use abbreviation.
Corrected (lines 173, 395-96, 519- also including glucoronides abbreviations).
- Please add the reference for Part 2.2.2.
The reference was added (line 436).
- Line 240, please delete “is”.
Corrected (line 245).
- Line 488, “diferences” change to “differences”.
Corrected (line 496).
- Please check Line 411, “The details have been described in [80] and [37] and are briefly summarized here.”. the reference cited is right?
Yes, the references are the ones reported since both are review papers on mycotoxin biomarkers and therefore, include information on FB1.
Round 2
Reviewer 3 Report
the author responsed all the comments, and I agree with accept this manuscript